# Soil Microbial Indicators within Rotations and Tillage Systems

**DOI:** 10.3390/microorganisms9061244

**Published:** 2021-06-08

**Authors:** Gevan D. Behnke, Nakian Kim, Maria C. Zabaloy, Chance W. Riggins, Sandra Rodriguez-Zas, Maria B. Villamil

**Affiliations:** 1Department of Crop Sciences, University of Illinois, Urbana, IL 61801, USA; gbehnke2@illinois.edu (G.D.B.); nakhyun2@illinois.edu (N.K.); cwriggin@illinois.edu (C.W.R.); 2Centro de Recursos Naturales Renovables de la Zona Semiárida (CERZOS, UNS-CONICET), Departamento de Agronomía, Universidad Nacional del Sur, Bahia Blanca B8000, Argentina; mzabaloy@uns.edu.ar; 3Department of Animal Sciences, University of Illinois, Urbana, IL 61801, USA; rodrgzzs@illinois.edu

**Keywords:** bacteria, fungi, archaea, metagenomics, microbial N cycle, nitrification, maize, soybean, monocultures, no tillage

## Abstract

Recent advancements in agricultural metagenomics allow for characterizing microbial indicators of soil health brought on by changes in management decisions, which ultimately affect the soil environment. Field-scale studies investigating the microbial taxa from agricultural experiments are sparse, with none investigating the long-term effect of crop rotation and tillage on microbial indicator species. Therefore, our goal was to determine the effect of rotations (continuous corn, CCC; continuous soybean, SSS; and each phase of a corn-soybean rotation, Cs and Sc) and tillage (no-till, NT; and chisel tillage, T) on the soil microbial community composition following 20 years of management. We found that crop rotation and tillage influence the soil environment by altering key soil properties, such as pH and soil organic matter (SOM). Monoculture corn lowered pH compared to SSS (5.9 vs. 6.9, respectively) but increased SOM (5.4% vs. 4.6%, respectively). Bacterial indicator microbes were categorized into two groups: SOM dependent and acidophile vs. N adverse and neutrophile. Fungi preferred the CCC rotation, characterized by low pH. Archaeal indicators were mainly ammonia oxidizers with species occupying niches at contrasting pHs. Numerous indicator microbes are involved with N cycling due to the fertilizer-rich environment, prone to aquatic or gaseous losses.

## 1. Introduction

Agricultural management practices influence soil microbial communities, creating niche environments that favor certain microbes [1,2]. Management practices can include crop rotation, tillage, N fertilization, cover cropping, etc. By selecting management practices or combining them, the soil environment is altered, as are essential soil processes. These can include residue decomposition, nutrient and water cycling, aeration and gaseous interactions, development of soil aggregates, soil organic matter (SOM) dynamics, and biodiversity measures [3,4]. Crop rotation is a common management practice with benefits that include pest and disease control and yield improvement and stabilization [5,6,7]. Tillage is another tool used to improve yields by creating a more favorable environment for cash crop growth. In systems of high organic matter, tillage ensures a clean seedbed for early growth by reducing compaction, improving aeration, increasing soil temperature, and removing weed competition [5,8,9]. Lastly, N fertilization is a common practice used to enhance yields, and that influx of previously scarce N reshapes potential N dynamics controlled by soil microbial communities [10].

Given the benefits of crop rotation, tillage, and N fertilization on crop yields, their implementation is widespread, which affects the soil microbial community. Crop rotation and fertilization alter the quantity and quality of crop residues, root exudates, and subsequent rhizodeposits [11,12,13]. Results from a meta-analysis by Ouyang et al. [14] showed that crop rotation and soil pH influenced N cycling by changing the ammonia-oxidizing bacteria (AOB) and archaea (AOA) community dynamics, as well as denitrifiers. Crop rotation increased AOB levels compared to monocultures; AOA was unaffected. However, neutrophilic soil conditions led to an increase in both AOA and AOB. Furthermore, N fertilization increased the abundance of AOA, AOB, and denitrifiers [14]. In a remarkably long-term study on the Morrow Plots in Urbana, IL, treatments of crop rotation and N fertilization have been in place since 1876; shifts in microbial functions related to substrate utilization were affected by fertilizer treatments more than crop rotation given the chronic nutrient limitations and changes in soil chemistry [15]. Smith et al. [16] reported that crop type and tillage altered species composition, but not quantity and diversity metrics from Indiana corn (*Zea mays* L.)–soybean [*Glycine max* (L.) Merr.] rotations. Compared to crop rotation, tillage had a larger effect on nutrient levels, which was a better predictor for microbial community composition [16]. The effects of tillage on the soil microbial community include mechanically disrupting growth and distribution, destroying soil aggregates, reducing soil moisture, increasing soil temperature, and degrading soil organic matter (SOM) [17,18,19]. In a global meta-analysis looking at the effects of tillage, Zuber and Villamil [20] observed that NT systems have greater microbial biomass and enzymatic activities. Furthermore, de Graaff et al. [21], also using a meta-analytic approach, showed that tillage decreased bacterial biodiversity, however, it did not affect fungi.

Previously when technology was a limiting factor, using broad inference measurements was the best available technique for explaining how the soil microbiome responds to management factors. However, new metagenomic approaches better characterize the microbial community composition and function and its relationship with soil properties and agronomics [22,23,24]. Diversity and richness metrics represent the variability within a single sample (α-diversity) and among communities (β-diversity). Using quantitative polymerase chain reaction (qPCR), functional microbial genes, such as nirK, which is involved in denitrification, are analyzed for treatment effects [10,25]. Lastly, using primers for each major taxonomic group (bacteria, fungi, and archaea, PCR amplification produces a vast pool of amplicons. From that pool of hundreds to thousands of individual amplicons, high throughput sequencing with Illumina yields a deep inventory of amplicon sequence variants (ASVs), from where indicator microbes can be selected and characterized [22,23]. Indicator microbes usually refer to an ASV that explains variability in a dataset [23]. Studies on indicator microbes have shown that organic matter inputs and pH alter the cycling of N and C, resulting in significant changes in soil biological properties [2,15,26,27]. Given the complexity of using metagenomics to identify indicator microbes, field studies are scarce, especially from a long-term setting. A few long-term studies (15–130 years) have determined indicator microbes from typical cropping systems [2,15,27], though none have analyzed crop rotation and tillage simultaneously. As these are the most common tools used by growers to improve yields, a thorough investigation of these indicator microbes is necessary.

We hypothesized that our treatments of continuous corn and soybean would show contrasting effects on microbial taxa, with rotated corn-soybean having intermediate results, not different from either monoculture. We also hypothesized that AOB and fungi would have elevated abundances in the continuous corn treatment, with AOA increasing in the continuous soybean treatment. Therefore, the objective of this investigation was to identify microbial taxa that were responsive to crop rotation and tillage from a long-term, stable trial (20+ years). The results will add valuable primary information on how the soil microorganisms shift in response to common agricultural management practices.

## 2. Materials and Methods

### 2.1. Experimental Site Description and Management Practices

The experiment was conducted at the Northwestern Illinois Agricultural Research and Demonstration Center (40°55′50″ N, 90°43′38″ W), near Monmouth, Illinois. The study was established in 1996, and a complete description of the site can be found in Behnke et al. (2018; 2020). Briefly, soils were comprised of highly fertile silty clay loam and silt loam soil series (Muscatune 43%, Sable 40%, and Osco 17%) [28]. The study was designed in a split-plot arrangement of 4 rotation levels and 2 tillage levels in a randomized complete block design with 4 replications (blocks). The main plots (22 m long by 12 m wide) were crop rotation treatments, which consisted of continuous corn (CCC), corn phase of the corn-soybean rotation (Cs), soybean phase of the corn-soybean rotation (Sc), and continuous soybean (SSS). Subplot (22 m long by 6 m wide) tillage options were either no-till (NT) or chisel tillage (T). Tillage occurred in the fall following harvest using a disk-ripper to 35 cm in depth, and in the spring, a soil finisher was used to prepare the seedbeds. No-till plots received zero tillage. Spring N fertilizer was applied at or before planting as injected incorporated urea ammonium nitrate (UAN) at a rate of 246 kg N ha^−1^ for CCC and 202 kg N ha^−1^ for Cs; soybeans received no fertilizer. Fertilizer and pest management followed the Illinois Agronomy Handbook [29]. Key field event date ranges are provided in Appendix A.

### 2.2. Soil Sampling and Procedures

Soil samples were taken postharvest in October of 2015 and 2016 using an Eijkelkamp grass plot sampler (Eijkelkamp Soil and Water, Morrisville, NC, USA) to a depth of 10 cm. A total of 3 subsamples were taken for each plot; each subsample consisted of around 10 random plugs totaling ~500 g of soil per subsample. A complete, multivariate examination of soil properties was conducted in Behnke, Zabaloy, Riggins, Rodriguez-Zas, Huang and Villamil [25], and a table summarizing that information was provided in the supplementary information Appendix A. Soil samples were immediately preserved with ice then frozen to -20 °C after returning to the lab facilities. Using 0.25 g of soil per composited subsample, soil DNA was extracted using the PowerSoil^®^ DNA isolation kits (MoBio Inc., Carlsbad, CA, USA), following the included instructions. The extracted DNA was then measured for quantity and quality using a Nanodrop 100 Spectrophotometer (Thermo Fisher Scientific, Waltham, MA, USA) and stored at −20 °C. Amplification of the Bacterial 16S rRNA gene (V4 region) used a primer set of 515F (GTGYCAGCMGCCGCGGTAA) and 806R (GGACTACVSGGGTWTCTAAT) [26], archaeal 16S used 349F (GTGCASCAGKCGMGAAW) and 806R (GGACTACVSGGGTATCTAAT) [30], and fungal ITS (internal transcribed spacer) region used 3F (GCATCGATGAAGAACGCAGC) and 4R (TCCTCCGCTTATTGATATGC) [31]. The primers were designed as a 50-PCR-specific + gene region + 30-PCR-specific + 10 nt barcode, and the Fluidigm platform used 2 primer sets concurrently in the creation of the final DNA amplicon. A Qubit Fluorometer (Thermo Fisher Scientific, Waltham, MA, USA) quantified the resulting amplicon libraries, which were then computed using a Bioanalyzer (Agilent, Santa Clara, CA, USA) to evaluate the profile of fragment lengths. The barcoded libraries were combined in equimolar concentrations and diluted to 10 nM. The diluted libraries were sequenced at the Roy Carver Biotechnology Center Functional Genomics lab at the University of Illinois at Urbana-Champaign (Urbana, IL, USA) using paired-end sequencing on the Illumina HiSeq (Illumina, San Diego, CA, USA), resulting in 250 nt long reads.

### 2.3. Bioinformatics Analysis

Using QIIME2 [32,33], the sequences were processed and checked for quality. Next, the demultiplexed sequences were filtered using a Q score threshold of 30 [34], which resulted in the retention of bacterial sequences between base-pair positions 6 to 231, fungal sequences 6 to 222, and archaeal sequences 6 to 221. Then, chimeric and low-quality sequences were removed by the denoising option (chimera-method consensus) in the plugin DADA2 [35]. The product of those steps was ASVs, which were then aligned with MAFFT (v7) [36] to generate the phylogenetic tree using FastTree [37] for β-diversity measurements. In Qiime 2.0, the option feature-classifier classify-sklearn was used to classify the ASVs with reference sequences in the SILVA ribosomal RNA gene database (silva-132-99-515-806-nb-classifier_2019_4) (Quast et al., 2013) and Fungi_97_classifier_2019_4. The rarefaction curves plateaued at sampling depths of 35,100 bacterial sequences per sample, 10,000 fungal sequences per sample, and 1000 archaeal sequences per sample (Appendix A). Using these depths, QIIME2 produced the number of observed ASVs, Pielou’s Evenness Index, and Shannon’s Diversity Index (H’) for each sample (shown in Table 1). Similarly, QIIME2 calculated β-diversity measurements for each taxa using weighted UniFrac distances (Table 2, Table 3 and Table 4). The rarefaction curves (Appendix A) were produced using SigmaPlot (v. 12.5 Systat Software, Inc., San Jose, CA, USA).

### 2.4. Statistical Analysis

In order to identify the responsive microbes and estimate treatment effects, the relative abundances (RAs, %) of each ASV were examined. [As described in Kim, Zabaloy, Riggins, Rodríguez-Zas, and Villamil [23]] Using the JMP^®^ predictor screening platform based on bootstrap forest partitioning [38,39] on the original dataset, a condensed set of responsive microbes was produced. The responsive dataset consisted of 35 out of 4098 bacterial ASVs, 37 out of 390 fungal ASVs, and 11 out of 28 archaeal ASVs. These responsive microbes contributed to at least 1% of the variability in the model algorithms (Appendix A). Then, the responsive microbe ASVs for each taxon were analyzed by principal component analysis (PCA) to further remove redundancy and avoid multicollinearity issues. Next, the RAs of these ASVs were summarized into a set of uncorrelated, orthogonal

Bacterial RAs for the phyla level showed that Proteobacteria (30%) was the most abundant, followed by Acidobacteria (20%), Actinobacteria (17%), Chloroflexi (10%), Planctomycetes (10%), Bacteroidetes (7%), Rokubacteria (3%), and Verrucomicrobia (3%) variables called principal components (PCs) utilizing the FACTOR procedure in SAS (v 9.4 SAS Institute, Cary, NC). The PCs with eigenvalues ≥1 that explained >5% of the variability in the dataset were then used as independent variables for further analysis. The ASVs with PC loading values > |0.5| were considered significant and classified as responsive microbes [40]. Using the GLIMMIX procedure in SAS, linear mixed models were fit to each responsive variable, including the α-diversity measures (Table 1) and PC scores of top contributing ASVs. Crop rotation and tillage were considered fixed effects, whereas blocks and years were considered random terms in the analyses of variance. Using SAS, least-square means of the response variables were separated by treatment levels and interactions, using the *pdiff* option and setting the probability of a type I error at α = 0.05. SigmaPlot was used to visualize the RA responses for each significant effect between the PC scores and treatments. The figures presented in the results characterized the combined PCA results and associated means separations for indicator taxa ASVs based on their RAs (the complete list of indicator microbes within each PC by taxa is shown in Appendix A). The calculations displayed in those figures were the mean PC score for a given treatment multiplied by the PC loading score for the listed ASVs; error bars represented the standard error of the mean for each PC score by treatment multiplied by the absolute value of each ASV loading. Calculations for β-diversity, which used weighted UniFrac distance, were conducted by QIIME2 using pairwise PERMANOVA (permutational multivariate analysis of variance) for comparing differences between treatment levels by pseudo-F test statistics and their *p*- and q-values, which represented the expected false positive (p) and negative rates (q) in multiple hypothesis testing [41,42].

## 3. Results

### 3.1. Overall Characterization of Soil Indicator Microbes

The bacterial kingdom had 47,888,681 16S V4 region sequences clustered into 4098 ASVs. The fungal kingdom had 5,253,422 ITS region sequences clustered into 390 ASVs. The archaeal kingdom had 2,380,099 archaeal 16S rRNA region sequences clustered into 28 ASVs. The α-diversity measurements (ASV counts, Pielou, and H’) for bacteria and archaea revealed no statistical differences for the main effects of crop rotation and tillage (Table 1). However, fungi showed CCC having statistically greater ASV counts and H’ measurements (*p* = 0.01 and 0.03, respectively) compared to SSS, with the rotated corn and soybean treatments being not different from either monoculture.

The analysis for β-diversity in the bacterial kingdom structure differed significantly (*p* < 0.01) for 26 out of the 28 rotation by tillage interactions, the exceptions being CsNT-ScNT and CsT-ScT (Table 2). The β-diversity in the fungal kingdom structure differed significantly for 13 out of the 28 treatment interactions, most of which were comparisons between corn and soybean phases (Table 3). The β-diversity measurements for the archaeal domain structure differed significantly for 9 out of the 28 treatment interactions, with 8 of the 9 comparisons driven by the monocultures (Table 4).

(Appendix A). The fungal community was dominated completely by the phylum Ascomycota (100%), of which seven out of the eight indicator species came from the class Sordariomycetes (Appendix A). The indicator species for the archaeal community were nearly all from the phylum Thaumarchaeota (73%), followed by Euryarchaeota (18%) and Nanoarchaeaeota (9%) (Appendix A).

### 3.2. ASVs Responses to Crop Rotation and Tillage Treatments

#### 3.2.1. Bacteria

The PCA on the 16S V4 bacterial domain produced five PCs (PC1-PC5; Table 5 and Appendix A), explaining 65% of the variability in the 35 selected top-contributing ASVs. Within each PC, bacterial indicator microbes were flagged when significant correlations (loadings ≥1 |0.5|) were detected. The taxonomic classification of these ASVs provided by the SILVA database is listed in Appendix A.

PC1 explained 36% of the variability, which contained significant loadings from 23 ASVs by the most explanatory taxonomic rank (listed in parenthesis: P, phylum; C, class; O, order; F, family; G, genus; S, species) (Appendix A). Positive loadings for PC1 were detected from *Paludibaculum* (G), Pyrinomonadaceae (F), Holophagae (C), Acidobacteria (P), Actinomarinales (O), Actinomycetales (O), Actinobacterium (O), Anaerolineae (C), Dehalococcoidia (C), Planctomycetes (P), Phycisphaerae (C), *Nordella* (G), Betaproteobacteriales (O), and Rokubacteriales (O). Negative loadings from PC1 were from Acidobacteriaceae (Subgroup 1) (F), *Candidatus Solibacter* (S), Microbacteriaceae (F), Ktedonobacterales (O), Tepidisphaerales (O), Micropepsaceae (F), *Chujaibacter* (G), *Rhodanobacter* (G), and Pedosphaerales (O). PC2 explained 9% of the variability and saw only two negative loadings from Chitinophagales (O) and Nitrosomonadaceae (F). PC3 explained 8% of the variability and had positive loadings from Sphingobacteriales (O) and Gammaproteobacteria (C) and one negative loading from *Gaiella* (G). PC4 explained 6% of the variability and contained one positive loading from Archangiaceae (F). PC5 explained 6% of the variability and contained one negative loading from *Luteimonas* (G).

The results from the bacterial ANOVA (Table 5) detected significant main effects for crop rotation and tillage (*p* = 0.0001) from PC1 (Figure 1) and just tillage effects for PC4 (*p* = 0.0041) and PC5 (*p* = 0.0001) (Figure 2). In PC1, the means separation procedure showed that treatment mean PC scores from SSS were significantly greater than the other three treatments and rotated corn and soybean being greater than CCC. In PC4, the means for tillage were found to be greater than no-till. PC2 (*p* = 0.006) and PC3 (*p* = 0.0142) both saw a significant interaction between crop rotation and tillage (Figure 2). The interaction in PC2 showed an intricate interaction with SSST having the greatest mean but not different from CCCNT and CCCT; the rotated treatments were not different and were generally the lowest. The interaction for PC3, however, was more pronounced, with CCCT having the largest mean and all of the combinations being significantly lower but not different from each other (Figure 2).

#### 3.2.2. Fungi

The PCA on the ITS fungal kingdom produced 5 PCs (PC1-PC5; Table 5 and Appendix A), explaining a total of 34% of the variability in the 37 selected top-contributing fungal ASVs. As with bacteria, within each PC, fungal indicator species were flagged when significant correlations (loadings ≥ |0.5|) were found and identified to the nearest classification as provided by the SILVA database; fungal classification specifics are listed in Appendix A.

PC1 explained 8% of the variability and contained one significant positive loading from the ASV *Fusarium sporotrichioides* (S). PC2 explained 7% of the variability and contained positive loadings from Mycosphaerellaceae (F), *Gibellulopsis piscis* (S), and *Plectosphaerella* (G). PC3 also explained 7% of the variability and contained positive loadings from Coniochaetaceae (F), *Schizothecium* (G), and *Schizothecium carpinicola* (S). PC4 explained 6% of the variability and contained a positive loading from *Clonostachys rosea* (S). PC5 explained 6% of the variability, however, no significant correlations (loadings ≥ |0.5|) were selected.

The fungal ANOVA results (Table 5) found a significant crop rotation effect (*p* = 0.0163) for PC1 and a tillage effect for PC5 (*p* = 0.0348) (Figure 3). The means separation procedure for PC1 showed that CCC was the largest, the rotated treatments in the middle, and SSS having the lowest mean. The mean for NT was significantly lower than the mean for till in PC5, however, no indicator species was selected and thus will not be discussed further. A significant interaction between crop rotation and tillage (*p* = 0.0416) was detected for PC3. The interaction for PC3 was likely driven by a highly significant response of tillage, showing no-till being lower than till. This is confirmed by the greatest mean values occurring in CCCT, CsT, and ScT; the SSST treatment was not different from the NT pairs except for the CCCNT, which was the lowest overall (Figure 3). PC2 and PC4 contained no significant ANOVA findings and, therefore, will not be discussed further.

#### 3.2.3. Archaea

The PCA on the 16S rRNA archaeal domain produced 5 PCs (PC1-PC5; Table 5 and Appendix A), explaining a total of 69% of the variability in the 11 selected top-contributing archaeal ASVs. As with bacteria and fungi, archaeal indicator microbes were flagged when significant correlations (loadings ≥ 1 |0.5|) were discovered and identified by the SILVA database to the nearest classification; archaeal classification specifics are listed in Appendix A.

PC1 explained 20% of the variability and contained positive loadings from *Candidatus Nitrocosmicus* (S), Nitrososphaeraceae (F) (ammonia oxidizer), Nitrososphaeraceae (F); negative loadings came from Nitrososphaeraceae (F) and *Candidatus Nitrosotalea* (S). PC2 explained 15% of the variability and contained negative loadings from two different Nitrososphaeraceae (F) families. PC3 explained 12% of the variability and contained one positive loading from Woesearchaeia (C). PC4 explained 11% of the variability and also contained one positive loading from Thermoplasmata (C). PC5 explained 10% of the variability and contained opposing loadings, one positive from Thermoplasmata (C) and one negative from *Candidatus Nitrososphaera* (S).

The results from the ANOVA on the archaeal community found a crop rotation effect (*p* = 0.008) for PC1 (Figure 4). The means separation procedure showed that SSS was the largest but not different from Cs, which was not different from Sc; the CCC rotation was the lowest (Figure 4). Significant tillage main effects were detected for PC3 (*p* = 0.045) and PC4 (*p* = 0.0032) with tillage being greater than no-till in both cases (Figure 4). PC2 and PC5 had no significant ANOVA findings and, therefore, will not be discussed further.

## 4. Discussion

Overall, the results from this study indicate that crop rotation and tillage affect soil microbial guilds significantly. The monocultures of corn and soybean had contrasting effects on microbial taxa, with the rotated crops showing intermediate effects. No-till and tillage also had contrasting effects, and like crop rotation, could be used to explain microbial indicators. Other studies also showed that CCC and SSS have contrasting effects on the soil indicator microbes, and the CS rotation having similar effects on both monocultures [1,2]. Crop rotation affects the quantity and quality of plant residues, which are the food source for microbes, resulting in functional changes performed by soil microbes; likewise, monocultures or crops requiring extensive pesticides affect microbial diversity and richness [1]. Chamberlain, Bolton, Cox, Suen, Conley, and Ané [2], too, found that crop residues were likely the driving factor in the community shifts between CCC and SSS, indicating that the quality of organic matter should be considered. Soman, Li, Wander, and Kent [15] observed that long-term crop rotation shifted microbial taxa into distinct communities based on the rotation. However, since the Morrow Plots are an extreme example of nutrient deficiencies, plots receiving manure or inorganic fertilizer generate more residues, leading to greater SOM and enhanced microbial diversity. Similarly, NT is likely to increase bacterial diversity compared to tilled systems by increasing SOM [21]; NT also enhances microbial biomass and enzymatic activity by creating a favorable microclimate [20]. Soil pH is also an important factor that affects community structure, which can be altered by fertilization [43,44]. Therefore, cropping systems, rotation or tillage, that significantly alter SOM and pH could be used to classify indicator microbes.

The bacterial indicator microbes were categorized from seven different phyla and grouped by their responses to the treatments presented in this study. The CCC rotation is characterized as having high SOM due to large amounts of biomass returned annually, low soil pH due to N fertilization, elevated levels of *nirK* denitrification, and significantly greater levels of ammonia-oxidizing bacteria [25]. Thus in terms of N cycling, the CCC rotation was the most intense. The low soil pH was likely caused by the increased fertilizer rates compared to rotated corn (246 kg N ha^−1^ vs. 202 kg N ha^−1^) and receiving fertilizer yearly. Chamberlain, Bolton, Cox, Suen, Conley, and Ané [2] detected that soil pH and SOM explained the most variation, and like in our study, found that soil pH was lowest in the spring for CCC and SOM was the greatest from the CCC year-round. Ashworth, DeBruyn, Allen, Radosevich and Owens [1] also found that CCC had greater levels of C and N compared to a CS or SSS rotation and concluded that the inclusion of soybean into a crop rotation depletes soil organic C, which is a favored food source for microbes. Corn in general leaves behind about three times more residue than soybean following harvest [45] and has a higher C:N ratio, making the residues take longer to decompose [46]. In our study, the NT treatment responded similarly to CCC (Figure 1). Expectedly, the RAs of indicator microbes showed similar responses to both crop rotation and tillage, where those increased with CCC also increased with NT. Given the use of SOM and pH to accurately identify treatment effects, we have grouped our bacterial indicator microbes in two groups: CCC and NT vs. SSS and T.

Thus, we have identified 12 bacterial ASVs with increased RAs from CCC and NT, or high SOM and low pH associated microbes, including *Rhodanobacter* (G), *Chujaibacter* (G), Microbacteriaceae (F), Chitinophagales (O), Nitrosomonadaceae (F), Acidobacteriaceae (F), *C. solibacter* (S), Pedosphaeraceae (F), Micropepsaceae (F), Tepidisphaerales (O), Ktedonobacterales (O), and *Gaiella* (G) (Appendix A). *Rhodanobacter* (G) and *Chujaibacter* (G) are both members of Rhodanobacteraceae (F) and have been found to be the second most unaffected by agricultural use [first being the *Nitrosospira* (G)] and negatively correlated to soil pH [24]. *Rhodanbacter* (G) was found to amplify the *nirK* denitrification gene [47,48]. Green, Prakash, Gihring, Akob, Jasrotia, Jardine, Watson, Brown, Palumbo, and Kostka [48] found that *Rhodanobacter* (G) isolates accumulated nitrous oxide (N_2_O) during denitrification under their growing conditions; however, *nosZ* gene amplification was not observed. The relationship between increased *nirK* and accumulation of N_2_O likely translates into increased N_2_O emissions. This supposition was supported by Behnke, Zuber, Pittelkow, Nafziger, and Villamil [5], who showed that the CCC-T rotation by tillage combination, also taken from this study during 2012–2015, emits the most N_2_O compared to the other rotation by tillage combinations tested. Microbacteriaceae (F) are known to thrive under conditions of urea amendments and decreased pH, which is explained by elevated RAs from the CCC rotation (Figure 1) and observed in Staley et al. [49] using microcosms of soil from corn and soybean trials comparing tillage and N rate. Chitinophagales (O) is responsible for degrading organic matter [50] and is also associated with urea amendments and low pH [49]. Urea applications also increased the RAs for Nitrosomonadaceae (F) [49]. The low soil pH likely increased the RAs for Acidobacteriaceae (F) (Figure 1), which were enhanced in the CCC rotation. Other studies have observed increases in Acidobacteriaceae (F) as soil pH decreases [51,52,53]. Wang, et al. [54] found that *C. solibacter* (S) was strongly associated with low pH and contaminated sites suggesting that it might hold an ecological niche in such systems. Ward et al. [55] found that Acidobacteria (P) play a significant role in terrestrial C cycling, which would increase under chisel tillage. The CCC rotation, which also contains the largest levels of organic matter and crop residues [25], increased RAs from *C. solibacter* (S), which has been linked to increased levels of rice straw residue returned to the system and identified as a key species in the understanding of ecological processes from fertilized agroecosystems [56]. Pedosphaeraceae (F) have little information available but have been found in the soil and are thought to be associated with organic matter decomposition as they showed a positive correlation with Solibacteraceae (F) [57]. Micropepsaceae (F) information is scarce but has been found in acidic soils and is involved with carbon cycling [58]. Tepidisphaerales (O) has little information available except for an association with conventional farming [59], and WD2101(F) is polysaccharide degraders found in raised bogs and eutrophic fens [60]. Similarly, Ktedonobacterales (O) is relatively unknown but have been found to inhabit forests, gardens, and sand in low numbers, as well as extreme environments, such as volcanoes and geothermal areas [61]. However, Neupane, Bulbul, Wang, Lehman, Nafziger, and Marzano [27] detected elevated Ktedonobacterales (O) levels from a CCC rotation, although little supporting information regarding their biological significance was found. The increased RAs from the CCC rotation and chisel tillage show that these resilient microbes are adapted to the harsh and acidic conditions in this treatment. Little is known about *Gaiella* (G), aside from having a suspected relationship with plants and C cycling traits similar to related species [62,63]. In general, the bacterial ASVs associated with CCC and NT, likely favor nutrient-rich environments and are copiotrophic.

Conversely, the SSS rotation is characterized as having low SOM due to high C:N ratio biomass and much less, in terms of amounts, compared to corn, closer to neutral soil pH due to no N fertilization, decreased levels of *nirK* denitrification, and significantly lower levels of ammonia-oxidizing bacteria [25] (Appendix A). The chisel tillage treatment also has reduced levels of SOM due to quicker decomposition rates as tillage helps to increase residue surface area; likewise, chisel tillage incorporates fertilizer urea (in our case), leading to less surface soil acidification, as is the opposite in the NT treatment [25] (Appendix A).

The 18 bacteria with increased RAs from SSS and T are grouped into what we are calling neutral pH and reduced fertilizer N associated microbes, including ASVs belonging to Rokubacteriales (O), Phycisphaerae (C), Planctomycetes (P), Actinobacterium (O), Actinomycetales (O), *Luteimonas* (G), Holophagae (C), Acidobacteria (P), Pyrinomonadaceae (F), *Paludibaculum* (G), Sphingobacteriales (O), Archangiaceae (F), Betaproteobacteriales (O), *Nordella* (G), Dehalococcoidia (C), Anaerolineae (C), Actinomarinales (O), and Gammaproteobacteria (C). Rokubacteriales (O) RAs were increased in the SSS rotation (decreased in the CCC rotation) in our study, which is in line with another study showing decreased RAs from corn samples compared to woodland samples, likely due to contrasting nutrient levels dictating microbial guild specializations, even within the same phyla [64]. Only one study reported Phycisphaerae (C) results but found that increased RAs were associated with the lower SOM and N level treatment [65]. Planctomycetes (P), specifically OM190 (C), was found to be a dominant microbe in a system with wheat (*Triticum aestivum* L.) and corn straw returned, no fertilizer added; the organic C in this system was found to be significantly lower than the biomass + fertilizer treatments, however, soil pH was not different for any of the treatments but was near neutral (6.7–6.8) [66]. Actinobacterium (O) and Actinomycetales (O) are closely related and belong to MB-A2-108 (C) within Actinobacteria (P); these microbes are capable of degrading a wide range of organic material and are known to thrive in nutrient scarce environments [67]. Shange et al. [68] found that Actinomycetales (O) showed the largest RAs from soils with the lowest SOM values; Actinomycetales (O) is also an important member of the N cycling community [69]. *Luteimonas* (G) growth occurs from pH 5–9, with optimal growth at pH 7.0 [70]. Simmons, et al. [71] found *Luteimonas* (G) RAs were increased in unfertilized soil compared to soil amended with green waste compost. While Xiao et al. [72] found that *Luteimonas* (G) was positively correlated with soybean and alfalfa (*Medicago sativa* L.) biomass. Relatively little information exists for the Holophagae (C) Subgroup 7 (O), except that RAs were found to be positively correlated with a legume treatment compared to grass [73], with pH [74], and negatively correlated to SOM [75]. Acidobacteria (P) and many of their subgroup RAs were negatively correlated to nutrient levels and associated with C degradation [76]. Navarrete et al. [77] found that Subgroup 25 (C) was only found in soybean rhizospheric soil, not in their forest comparison site. Subgroup 25 (C) was also positively correlated to pH [78]. Pyrinomonadaceae (F) and *Paludibaculum* (G) both prefer only mildly acidic soils [79,80], which explains the RA increases under the SSS rotation. Likewise, Pyrinomonadaceae (F) prefers complex proteinaceous substrates [79], and *Paludibaculum* (G) is unable to utilize nitrate and urea [80] as would be the case in the SSS rotation. Multiple studies have found that Sphingobacteriales (O) RAs are negatively correlated to N fertilization [81,82,83]. Little information was available regarding Archangiaceae (F), but one study found that they are involved with the C cycle [84]. However, Myxococcales (O), the order for Archangiaceae (F), was found to be positively correlated with pH from agricultural lands, and RAs significantly increased from those under organic production [85]. Zhou et al. [86] also found a positive correlation between soil pH and Myxococcales (O) from forest and greenhouse soil. Furthermore, the authors discovered that bacterial community composition was a key factor in determining Myxococcales (O) RAs since they are predatory bacteria. Betaproteobacteriales (O) TRA3-20 (F) has no information available and could be misclassified. However, some members of the Gammaproteobacteria (C) are more abundant at neutral pH compared to acidic conditions [87]. *Nordella* (G), too, has little information available other than being more abundant at neutral to high soil pH [88]. Dehalococcoidia (C) are typically found in anaerobic conditions, but all grew best in neutral pH conditions with little growth in acidic conditions [89] and are known to be involved in C cycling [90]. Like Dehalococcoidia (C), Anaerolineae (C) is in the same phylum Chloroflexi and was also found to be associated with anaerobic conditions [91]. Yao et al. [92] found Anaerolineae (C) RAs declined as fertilizer N was introduced, though pH values were all above 8. Near neutral pH requirements were also observed by Yamada et al. [93] and Kandasamy, Weerasuriya, White, Patterson, and Lazarovits [88]. Cai et al. [94] recently discovered that Anaerolineae (C) were involved in the denitrification step of the N cycle from wastewater treatment sludge, though more research is needed to confirm such findings in an agricultural setting. In a German barley study comparing low and high N rates and mouldboard plow tillage compared to conservation tillage, Actinomarinales (O) RAs were increased under the low N setting using a mouldboard plow [95]. Much of the published research for Actinomarinales (O) has taken place in aquatic environments, and they likely play a significant role in the global C cycle [96]. Gammaproteobacteria (C) R7C24 (O) was likely misclassified as there is no information available except for a finding in a diabetes medical trial [97]. Ultimately, the bacterial ASVs associated with SSS and tilled systems likely favor nutrient-poor environments and are oligotrophic.

All indicator microbes for the fungal kingdom belonged to Sordariomycetes (C). *F. sporotrichioides* (S) was the dominant fungi identified through our PCA, explaining 8% of the variability by this single species. *F. sporotrichioides* (S) is a common, ineffectual agricultural and grassland pathogen [98], commonly isolated from corn and other cereal crops [99]. *F. sporotrichioides* (S) RAs have been reported to be negatively associated with soil pH [100], as is the case in this study showing greater RAs from the CCC rotation, which has significantly lower pH compared to the other rotations [25]. Behnke, Zabaloy, Riggins, Rodriguez-Zas, Huang and Villamil [25] also found that fungal ITS gene copy numbers were increased in the CCC rotation compared to SSS. Predictably, the CCC rotation observed significantly greater ASVs and H’ compared to SSS (Table 1). Tillage is also an important management factor that affects fungal RAs; Coniochaetales (O) Coniochaetaceae (F) and both ASVs from *Schizothecium* (G) are elevated from the tillage treatment as tillage is the driver of the significant interaction for PC3 (Figure 3). Larger Coniochaetales (O) RAs were observed in chisel-tilled wheat plots compared to NT [101] and also from the full tillage treatment from spring barley (*Hordeum vulgare* L.)–winter wheat–maize crop rotation [102]. Similarly, *Schizothecium* (G) RAs were increased in a standard tillage treatment compared to NT from a tomato (*Solanum lycopersicum* L.)-cotton (*Gossypium arboretum* L.) rotation [103] and also from a complex wheat–rapeseed (*Brassica napus* L.)–faba (*Vicia faba* L.) crop rotation [104]. Wang et al. [105] found that *Schizothecium* (G) RAs decreased in maize NT treatments. Sun et al. [106] studied three tillage treatments (moldboard plow, rotary tillage, and NT) and reported that fungal richness was significantly smaller in tilled systems compared to NT or reduced tillage. The increase of RAs in PC3 driven by tillage could indicate an adaption of these indicator fungi to disturbance, though more information would be needed to confirm.

The indicator microbes for archaea were taken from three classes of indicator microbes and grouped according to their responses to the treatments in the study. *C. Nitrosotalea* (S) and an uncultured Nitrososphaeraceae (F) RAs were greater in the CCC rotation. *C. Nitrosotalea* (S) is an ammonia oxidizer found in acidic agricultural soil [107,108,109,110], which is the case in our study as the CCC rotation has the lowest pH. Lehtovirta-Morley, et al. [111] and Papadopoulou, Bachtsevani, Lampronikou, Adamou, Katsaouni, Vasileiadis, Thion, Menkissoglu-Spiroudi, Nicol, and Karpouzas [110] both confirmed that *C. Nitrocosmicus* (S) occupy contrasting ecological niches compared to *C. Nitrosotalea* (S) and *C. Nitrocosmicus* is present in larger RAs at neutral pH. The unspecified AOA Nitrososphaeraceae (F) showing elevated RAs from the SSS rotation matches Behnke, Zabaloy, Riggins, Rodriguez-Zas, Huang, and Villamil [25]. The authors found increased AOA gene copy numbers in the SSS rotation, however, since the archaea are mostly uncultured, that cannot be fully confirmed. Yu, Lawrence, Sooksa-nguan, Smith, Tenesaca, Howe, and Hall [56] found that typically Nitrososphaeraceae (F) are AOA and positively associated with pH, but the pH relationship depended on niche specialties of certain archaea. Woesearchaeia (C) and Thermoplasmata (C) Marine Group II are typically found in aquatic ecosystems [112,113,114]. However, Wang et al. [115] found both archaea ASVs in agricultural and estuarial soils, noting that the agricultural soils contained significantly greater abundance, richness, evenness, and diversity than freshwater or estuarine ecosystems. There is no information regarding tillage practices that influence the RAs for these archaea. Nevertheless, the tillage treatment had significantly greater surface pH [25] (Appendix A), so pH could play a role in determining the RA for these two indicator archaea. Wang et al. [116] found that Woesearchaeia (C) was very abundant and negatively associated with salinity; while not a perfect relationship, typically the greater salinity, the lower the pH.

## 5. Conclusions

This study adds valuable insight as to important microbes and how they respond to typical agricultural management. We found that bacterial indicator microbes responded contrastingly to the two monocultures with the rotated corn and soybean showing intermediate effects, partially confirming our hypothesis. Tillage, too, showed contrasting effects between chisel tillage and NT. Using those strong main effects, we grouped the indicator bacteria into organic matter dependent and acidophile vs. N adverse and neutrophile. This grouping agreed with our previous qPCR publication [25] and fit bacterial characterizations well. From the indicator bacteria, we found that many were involved in the N cycle and respond positively to conditions of increased inorganic N. Unlike bacteria, fewer fungi and archaea were selected as indicator microbes. Fungi were poorly identified, and all were from Sordariomycetes (C), with the top indicator species thriving in the low pH environment of CCC, confirming our hypothesis. The significant archaeal indicators were mainly AOA, preferring the neutral pH in the SSS rotation, though some AOA were found to be acidophile, partially confirming our hypothesis. This study shows the need to fully classify soil organisms to a finer level, which will help better understand the role specific microorganisms play in soil nutrient cycling. Future work should focus on identifying the uncultured yet significant ASVs described in this study. In addition, N cycle genetic analysis of these indicator species would greatly help explain their role in the agroecosystems. Using metagenomics and bioinformatics, we were able to select 49 indicator microbes out of thousands of ASVs from highly productive soils, using typical agronomic management practices from a replicated, long-term trial. These indicator taxa could potentially generate a soil assessment narrative to identify inefficiencies in agronomic practices or indicate possible environmental consequences.

## Figures and Tables

**Figure 1 microorganisms-09-01244-f001:**
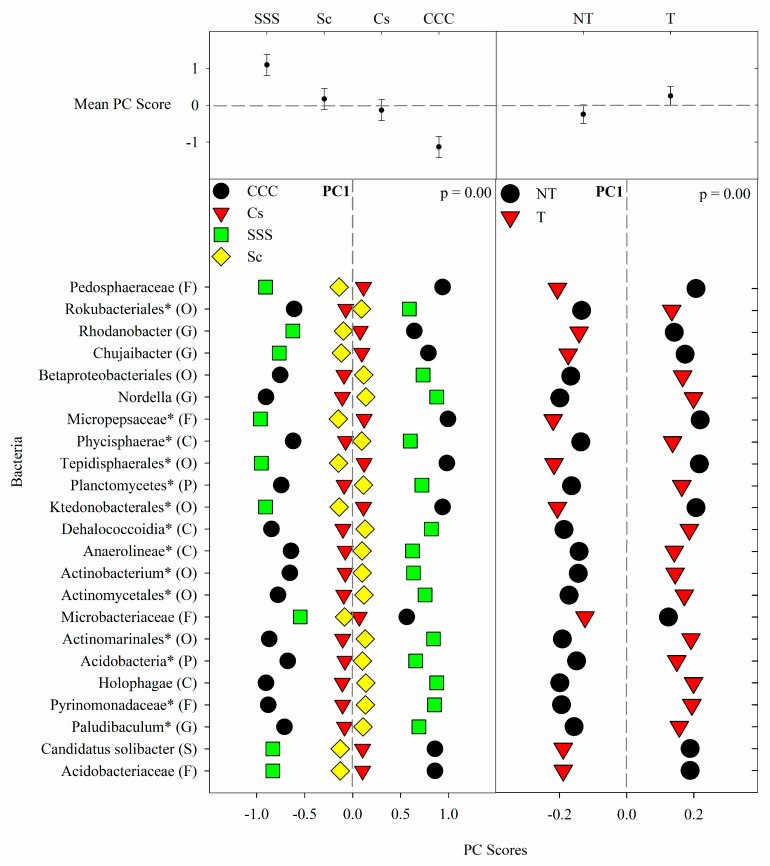
Mean bacterial principal component scores (PC) following 20 years of rotation and tillage treatments. Top panels show the bacterial mean PC score for both crop rotation and tillage main effects for PC1 based on the analysis of variance (ANOVA); error bars represent standard errors of the mean PC scores. Bottom panels show relative abundances (RAs) for each bacterial indicator ASVs by crop rotation and tillage effects. The main effects for PC1 are shown as CCC, continuous corn; Cs, corn phase of the corn-soybean rotation; Sc, soybean phase of the corn-soybean rotation; SSS, continuous soybean; NT, no-till; T, chisel tillage. For each taxon, the response of each ASV was calculated as the mean PC score multiplied by the PC loading score of a given ASV. The y-axes show the name of the ASV’s most explanatory taxonomic rank in parentheses (P, phylum; C, class; O, order; F, family; G, genus; S, species). The “*” after an ASV means it is uncultured.

**Figure 2 microorganisms-09-01244-f002:**
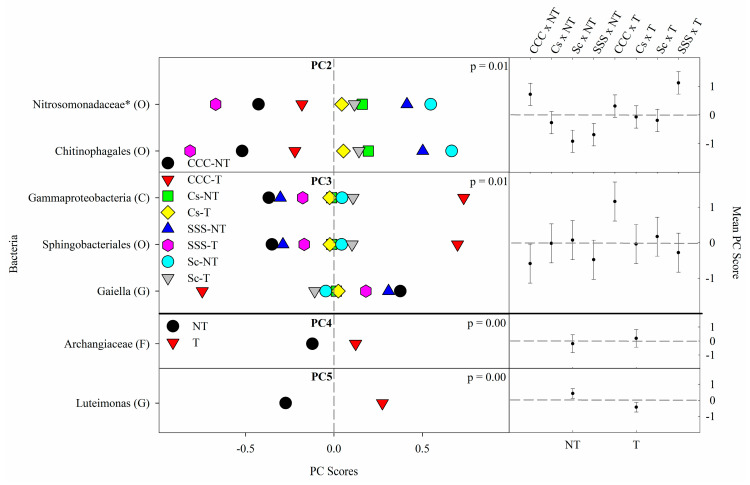
Mean bacterial principal component scores (PC) following 20 years of rotation and tillage treatments. Right panels show the bacterial mean PC score for the crop rotation × tillage interaction for PC2 and PC3 and a tillage effect for PC4 and PC5 based on the analysis of variance (ANOVA); error bars represent standard errors of the mean PC scores. Left panels show relative abundances (RAs) for each bacterial indicator ASVs by crop rotation × tillage and tillage. The crop rotation × tillage interaction for PC2 and PC3 is shown as CCC-NT, continuous corn, and no-till; CCC-T, continuous corn, and chisel tillage; Cs-NT, corn phase of the corn-soybean rotation and no-till; Cs-T, corn phase of the corn-soybean rotation and chisel tillage; Sc-NT, soybean phase of the corn-soybean rotation and no-till; Sc-T, soybean phase of the corn-soybean rotation and chisel tillage; SSS-NT, continuous soybean and no-till; SSS-T, continuous soybean and chisel tillage. The tillage main effect for PC4 and PC5 is shown as NT, no-till; T, chisel tillage. For each taxon, the response of each ASV was calculated as the mean PC score multiplied by the PC loading score of a given ASV. The y-axes show the name of the ASV’s most explanatory taxonomic rank in parentheses (P, phylum; C, class; O, order; F, family; G, genus; S, species). The “*” after an ASV means it is uncultured.

**Figure 3 microorganisms-09-01244-f003:**
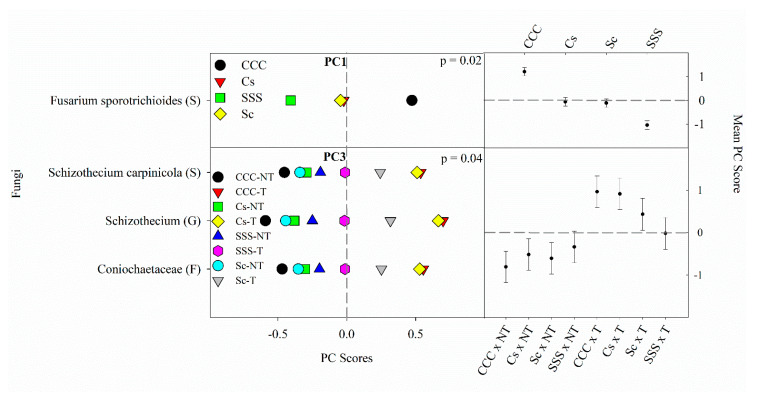
Mean fungal principal component scores (PC) following 20 years of rotation and tillage treatments. Right panels show the fungal mean PC score for the main effect of crop rotation for PC1 and the crop rotation × tillage interaction for PC3 based on the analysis of variance (ANOVA); error bars represent standard errors of the mean PC scores. Left panels show relative abundances (RAs) for each fungal indicator ASVs by crop rotation and crop rotation × tillage. The crop rotation main effects for PC1 are shown as CCC, continuous corn; Cs, corn phase of the corn-soybean rotation; Sc, soybean phase of the corn-soybean rotation; SSS, continuous soybean. The crop rotation × tillage interaction for PC3 is shown as CCC-NT, continuous corn and no-till; CCC-T, continuous corn, and chisel tillage; Cs-NT, corn phase of the corn-soybean rotation and no-till; Cs-T, corn phase of the corn-soybean rotation and chisel tillage; Sc-NT, soybean phase of the corn-soybean rotation and no-till; Sc-T, soybean phase of the corn-soybean rotation and chisel tillage; SSS-NT, continuous soybean and no-till; SSS-T, continuous soybean and chisel tillage. For each taxon, the response of each ASV was calculated as the mean PC score multiplied by the PC loading score of a given ASV. The y-axes show the name of the ASV’s most explanatory taxonomic rank in parentheses (P, phylum; C, class; O, order; F, family; G, genus; S, species).

**Figure 4 microorganisms-09-01244-f004:**
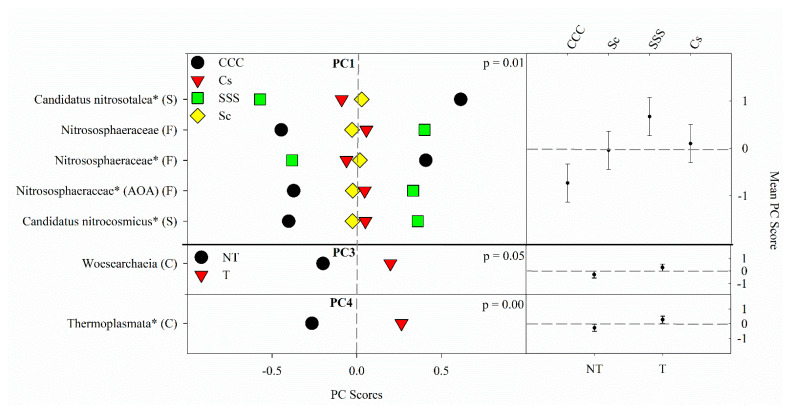
Mean archaeal principal component scores (PC) following 20 years of rotation and tillage treatments. Right panels show the archaeal mean PC score for the main effects of crop rotation for PC1 and tillage for PC3 and PC4 based on the analysis of variance (ANOVA); error bars represent standard errors of the mean PC scores. Left panels show relative abundances (RAs) for each archaeal indicator ASVs by crop rotation and tillage. The crop rotation main effects for PC1 are shown as CCC, continuous corn; Cs, corn phase of the corn-soybean rotation; Sc, soybean phase of the corn-soybean rotation; SSS, continuous soybean. The tillage effects for PC3 and PC4 are shown as NT, no-till; T, chisel tillage. For each taxon, the response of each ASV was calculated as the mean PC score multiplied by the PC loading score of a given ASV. The y-axes show the name of the ASV’s most explanatory taxonomic rank in parentheses (P, phylum; C, class; O, order; F, family; G, genus; S, species). The “*” after an ASV means it is uncultured. AOA denotes ammonia-oxidizing archaea.

**Table 1 microorganisms-09-01244-t001:** Mean values and standard errors of the mean (SEM) for the α-diversity parameters of observed amplicon sequence variants (ASVs), Pielou Evenness Index (Pielou), and Shannon’s Diversity Index (H’) for bacteria, fungi, and archaea taxa, following 20 years of crop rotation and tillage treatments. For each taxon group and within a given column, treatment mean values followed by the same lowercase letter were not statistically different (α = 0.05).

		ASVs			Pielou			H’		
Taxa	Treatment	Mean	SEM	*p*-Value	Mean	SEM	*p*-Value	Mean	SEM	*p*-Value
Bacteria	CCC ^†^	1297.08	62.29	0.42	0.97	0.00	0.99	10.00	0.07	0.30
	Cs	1377.79			0.97			10.12		
	Sc	1384.50			0.97			10.14		
	SSS	1445.19			0.97			10.20		
	NT ^‡^	1380.71	44.05	0.88	0.97	0.00	0.53	10.12	0.05	0.83
	T	1371.57			0.97			10.11		
Fungi	CCC	30.56 a	2.09	0.01	0.82	0.01	0.33	3.97 a	0.12	0.03
	Cs	29.50 ab			0.80			3.85 ab		
	Sc	25.17 ab			0.80			3.63 ab		
	SSS	23.90 b			0.79			3.54 b		
	NT	27.81	1.73	0.52	0.79	0.01	0.18	3.73	0.09	0.78
	T	26.75			0.81			3.77		
Archaea	CCC	64.34	6.22	0.80	0.93	0.00	0.51	5.54	0.14	0.83
	Cs	65.64			0.94			5.55		
	Sc	69.79			0.94			5.65		
	SSS	70.87			0.94			5.68		
	NT	68.81	4.86	0.68	0.94	0.00	0.85	5.63	5.63	0.69
	T	66.51			0.94			5.58		

^†^ CCC, continuous corn; Cs, corn phase of the corn-soybean rotation; Sc, soybean phase of the corn-soybean rotation; SSS, continuous soybean. ^‡^ NT, no-till; T, chisel tillage.

**Table 2 microorganisms-09-01244-t002:** Community structure (β-diversity) measures for bacteria following 20 years of crop rotation and tillage treatments, and their interaction, based on pairwise permanova computation of weighted unifrac distances. The pseudo-F column represents the comparison between UniFrac distances for a given treatment combination (listed in the Treatments Compared column). The *p*-value and q-value columns indicate the probability of type I and type II errors associated with the treatment comparisons, respectively.

Treatment	Treatments Compared	Sample Size	Pseudo-F	*p*-Value	q-Value
Rotation ^†^	CCC-Cs	96	3.97	0.0010	0.0012
	CCC-SSS	96	15.14	0.0010	0.0012
	CCC-Sc	96	5.35	0.0010	0.0012
	Cs-SSS	96	6.10	0.0010	0.0012
	Cs-Sc	96	1.30	0.1600	0.1600
	SSS-Sc	96	5.15	0.0010	0.0012
Tillage ^‡^	NT-T	192	5.45	0.0010	0.0010
Rotation × Tillage ^•^	CCCNT-CCCT	48	3.11	0.0030	0.0047
	CCCNT-CsNT	48	2.85	0.0040	0.0053
	CCCNT-CsT	48	5.54	0.0010	0.0020
	CCCNT-SSSNT	48	8.31	0.0010	0.0020
	CCCNT-SSST	48	11.37	0.0010	0.0020
	CCCNT-ScNT	48	3.12	0.0050	0.0064
	CCCNT-ScT	48	7.01	0.0010	0.0020
	CCCT-CsNT	48	1.94	0.0200	0.0224
	CCCT-CsT	48	2.71	0.0020	0.0035
	CCCT-SSSNT	48	7.57	0.0010	0.0020
	CCCT-SSST	48	9.49	0.0010	0.0020
	CCCT-ScNT	48	2.93	0.0010	0.0020
	CCCT-ScT	48	3.64	0.0010	0.0020
	CsNT-CsT	48	1.76	0.0310	0.0334
	CsNT-SSSNT	48	3.49	0.0020	0.0035
	CsNT-SSST	48	5.89	0.0010	0.0020
	CsNT-ScNT	48	0.85	0.6340	0.6340
	CsNT-ScT	48	2.02	0.0110	0.0128
	CsT-SSSNT	48	3.37	0.0010	0.0020
	CsT-SSST	48	3.57	0.0010	0.0020
	CsT-ScNT	48	2.68	0.0040	0.0053
	CsT-ScT	48	1.30	0.1400	0.1452
	SSSNT-SSST	48	2.07	0.0080	0.0097
	SSSNT-ScNT	48	3.04	0.0030	0.0047
	SSSNT-ScT	48	2.67	0.0010	0.0020
	SSST-ScNT	48	6.17	0.0010	0.0020
	SSST-ScT	48	3.20	0.0010	0.0020
	ScNT-ScT	48	2.43	0.0040	0.0053

^†^ CCC, continuous corn; Cs, corn phase of the corn-soybean rotation; Sc, soybean phase of the corn-soybean rotation; SSS, continuous soybean. ^‡^ NT, no-till; T, chisel tillage. ^•^ CCCNT, continuous corn, and no-till; CCCT, continuous corn, and chisel tillage; CsNT, corn phase of the corn-soybean rotation and no-till; CsT, corn phase of the corn-soybean rotation and chisel tillage; ScNT, soybean phase of the corn-soybean rotation and no-till; ScT, soybean phase of the corn-soybean rotation and chisel tillage; SSSNT, continuous soybean and no-till; SSST, continuous soybean and chisel tillage.

**Table 3 microorganisms-09-01244-t003:** Community structure (β-diversity) measures for fungi following 20 years of crop rotation and tillage treatments, and their interaction, based on pairwise permanova computation of weighted unifrac distances. The pseudo-F column represents the comparison between UniFrac distances for a given treatment combination (listed in the Treatments Compared column). The *p*-value and q-value columns indicate the probability of type I and type II errors associated with the treatment comparisons, respectively.

Treatment	Treatments Compared	Sample Size	Pseudo-F	*p*-Value	q-Value
Rotation ^†^	CCC-Cs	94	1.83	0.0270	0.0324
	CCC-SSS	95	3.68	0.0010	0.0060
	CCC-Sc	95	2.12	0.0150	0.0225
	Cs-SSS	95	2.21	0.0070	0.0200
	Cs-Sc	95	1.95	0.0100	0.0200
	SSS-Sc	96	1.35	0.1250	0.1250
Tillage ^‡^	NT-T	190	1.63	0.0450	0.0450
Rotation × Tillage ^•^	CCCNT-CCCT	47	1.25	0.1870	0.2277
	CCCNT-CsNT	48	1.26	0.1630	0.2075
	CCCNT-CsT	47	1.87	0.0300	0.0700
	CCCNT-SSSNT	48	2.61	0.0020	0.0280
	CCCNT-SSST	48	2.13	0.0280	0.0700
	CCCNT-ScNT	48	1.57	0.0780	0.1456
	CCCNT-ScT	48	1.85	0.0290	0.0700
	CCCT-CsNT	47	1.32	0.1310	0.1747
	CCCT-CsT	46	1.35	0.1210	0.1747
	CCCT-SSSNT	47	3.22	0.0010	0.0280
	CCCT-SSST	47	2.20	0.0120	0.0700
	CCCT-ScNT	47	1.60	0.0610	0.1220
	CCCT-ScT	47	1.70	0.0270	0.0700
	CsNT-CsT	47	0.88	0.5910	0.6129
	CsNT-SSSNT	48	1.69	0.0250	0.0700
	CsNT-SSST	48	1.41	0.1120	0.1742
	CsNT-ScNT	48	1.11	0.2710	0.3035
	CsNT-ScT	48	1.75	0.0210	0.0700
	CsT-SSSNT	47	1.89	0.0140	0.0700
	CsT-SSST	47	1.76	0.0390	0.0840
	CsT-ScNT	47	1.42	0.0950	0.1663
	CsT-ScT	47	1.82	0.0300	0.0700
	SSSNT-SSST	48	1.40	0.1260	0.1747
	SSSNT-ScNT	48	0.95	0.5040	0.5428
	SSSNT-ScT	48	2.34	0.0070	0.0653
	SSST-ScNT	48	0.78	0.7060	0.7060
	SSST-ScT	48	1.48	0.1050	0.1729
	ScNT-ScT	48	1.18	0.2230	0.2602

^†^CCC, continuous corn; Cs, corn phase of the corn-soybean rotation; Sc, soybean phase of the corn-soybean rotation; SSS, continuous soybean. ^‡^NT, no-till; T, chisel tillage. ^•^CCCNT, continuous corn, and no-till; CCCT, continuous corn, and chisel tillage; CsNT, corn phase of the corn-soybean rotation and no-till; CsT, corn phase of the corn-soybean rotation and chisel tillage; ScNT, soybean phase of the corn-soybean rotation and no-till; ScT, soybean phase of the corn-soybean rotation and chisel tillage; SSSNT, continuous soybean and no-till; SSST, continuous soybean and chisel tillage.

**Table 4 microorganisms-09-01244-t004:** Community structure (β-diversity) measures for archaea following 20 years of crop rotation and tillage treatments, and their interaction, based on pairwise permanova computation of weighted unifrac distances. The pseudo-F column represents the comparison between UniFrac distances for a given treatment combination (listed in the Treatments Compared column). The *p*-value and q-value columns indicate the probability of type I and type II errors associated with the treatment comparisons, respectively.

Treatment	Treatments Compared	Sample Size	Pseudo-F	*p*-Value	q-Value
Rotation ^†^	CCC-Cs	85	2.49	0.0620	0.1240
	CCC-SSS	89	7.08	0.0010	0.0060
	CCC-Sc	89	0.97	0.3270	0.3270
	Cs-SSS	88	1.56	0.1610	0.2415
	Cs-Sc	88	1.25	0.2320	0.2784
	SSS-Sc	92	5.24	0.0040	0.0120
Tillage ^‡^	NT-T	177	2.17	0.0920	0.0920
Rotation × Tillage ^•^	CCCNT-CCCT	43	1.20	0.2790	0.4595
	CCCNT-CsNT	41	1.02	0.3590	0.5026
	CCCNT-CsT	43	3.15	0.0310	0.1085
	CCCNT-SSSNT	44	3.49	0.0220	0.1085
	CCCNT-SSST	44	4.71	0.0030	0.0840
	CCCNT-ScNT	43	0.65	0.5930	0.6642
	CCCNT-ScT	45	1.73	0.1300	0.2595
	CCCT-CsNT	42	0.85	0.4010	0.5347
	CCCT-CsT	44	2.73	0.0520	0.1456
	CCCT-SSSNT	45	3.51	0.0270	0.1085
	CCCT-SSST	45	4.50	0.0130	0.1085
	CCCT-ScNT	44	0.80	0.4480	0.5702
	CCCT-ScT	46	1.03	0.3140	0.4884
	CsNT-CsT	42	1.71	0.1270	0.2595
	CsNT-SSSNT	43	1.83	0.1050	0.2450
	CsNT-SSST	43	2.78	0.0480	0.1456
	CsNT-ScNT	42	0.59	0.6570	0.7075
	CsNT-ScT	44	0.53	0.7680	0.7680
	CsT-SSSNT	45	0.55	0.7190	0.7456
	CsT-SSST	45	0.69	0.5480	0.6393
	CsT-ScNT	44	3.01	0.0310	0.1085
	CsT-ScT	46	1.19	0.2640	0.4595
	SSSNT-SSST	46	0.76	0.5050	0.6148
	SSSNT-ScNT	45	3.26	0.0230	0.1085
	SSSNT-ScT	47	1.76	0.1390	0.2595
	SSST-ScNT	45	4.95	0.0130	0.1085
	SSST-ScT	47	2.45	0.0610	0.1553
	ScNT-ScT	46	1.02	0.3450	0.5026

^†^ CCC, continuous corn; Cs, corn phase of the corn-soybean rotation; Sc, soybean phase of the corn-soybean rotation; SSS, continuous soybean. ^‡^ NT, no-till; T, chisel tillage. ^•^ CCCNT, continuous corn, and no-till; CCCT, continuous corn, and chisel tillage; CsNT, corn phase of the corn-soybean rotation and no-till; CsT, corn phase of the corn-soybean rotation and chisel tillage; ScNT, soybean phase of the corn-soybean rotation and no-till; ScT, soybean phase of the corn-soybean rotation and chisel tillage; SSSNT, continuous soybean and no-till; SSST, continuous soybean and chisel tillage.

**Table 5 microorganisms-09-01244-t005:** Analysis of variance (ANOVA) results for the effects of crop rotation, tillage, and their interaction (Rot × Till) on each group of principal components (PCs) calculated for bacteria, fungi, and archaea taxa datasets. The datasets for each taxa were comprised of indicator ASVs. The probability values (*p*-Value) for each treatment effect and degrees of freedom (df) are displayed in the top rows. The treatment mean values and their standard errors (SEM) are presented below. For each taxon group and within a given column, treatment mean values followed by the same lowercase letter were not statistically different (α = 0.05).

		Bacteria					Fungi					Archaea				
		PC1	PC2	PC3	PC4	PC5	PC1	PC2	PC3	PC4	PC5	PC1	PC2	PC3	PC4	PC5
Treatments	df	*p*-Value					*p*-Value					*p*-Value				
Rotation	3	0.00	0.01	0.18	0.29	0.19	0.02	0.42	0.47	0.83	0.61	0.01	0.20	0.98	0.95	0.39
Tillage	1	0.00	0.01	0.02	0.00	0.00	0.79	0.37	0.00	0.11	0.03	0.16	0.26	0.05	0.00	0.09
Rot × Till	3	0.31	0.01	0.01	0.68	0.06	0.09	0.79	0.04	0.70	0.66	0.15	0.39	0.79	0.21	0.55
Treatment means																
CCC ^†^		−1.13 a	0.51	0.29	−0.23	−0.01	1.21 a	−0.12	0.07	0.24	0.13	−0.73 c	0.09	0.02	−0.05	−0.52
Cs		−0.14 b	−1.69	−0.03	0.00	−0.61	−0.06 b	0.64	0.20	−0.29	0.07	0.10 ab	0.32	0.03	−0.08	0.30
Sc		0.17 b	−0.55	0.12	−0.29	0.52	−0.11 b	−0.39	−0.09	−0.02	−0.31	−0.04 b	−0.41	−0.12	0.06	0.11
SSS		1.09 c	0.21	−0.38	0.52	0.10	−1.04 c	−0.13	−0.18	0.07	0.11	0.67 a	0.00	0.07	0.07	0.11
SEM		0.29	0.33	0.52	0.68	0.37	0.18	0.41	0.33	0.42	0.42	0.40	0.34	0.38	0.29	0.36
NT ^‡^		−0.25 a	−0.29	−0.25	−0.19 a	0.43 a	−0.02	0.10	−0.57	0.19	−0.30 a	−0.11	−0.14	−0.28 a	−0.35 a	−0.19
T		0.25 b	0.29	0.25	0.19 b	−0.43 b	0.02	−0.10	0.57	−0.19	0.30 b	0.11	0.14	0.28 b	0.35 b	0.19
SEM		0.25	0.30	0.50	0.63	0.30	0.13	0.25	0.30	0.25	0.38	0.36	0.30	0.27	0.24	0.23
CCC-NT ^•^		−1.34	0.72 ab	−0.59 b	−0.42	0.85	0.97	0.04	−0.81 d	0.41	−0.29	−0.96	−0.03	−0.17	−0.81	−0.50
Cs-NT		−0.30	−0.27 cde	−0.02 b	−0.28	−0.31	0.02	0.86	−0.52 cd	−0.31	−0.16	0.16	0.52	−0.39	−0.18	−0.11
Sc-NT		−0.23	−0.92 e	0.07 b	−0.36	0.93	−0.27	−0.28	−0.61 cd	0.36	−0.37	−0.39	−0.78	−0.51	−0.20	−0.10
SSS-NT		0.87	−0.69 de	−0.48 b	0.28	0.27	−0.79	−0.21	−0.34 cd	0.29	−0.37	0.74	−0.25	−0.05	−0.23	−0.05
CCC-T		−0.91	0.31 abc	1.16 a	−0.04	−0.86	1.45	−0.29	0.96 a	0.06	0.56	−0.50	0.22	0.21	0.71	−0.55
Cs-T		0.03	−0.07 bcd	−0.04 b	0.28	−0.91	−0.13	0.43	0.91 a	−0.27	0.30	0.04	0.13	0.44	0.01	0.72
Sc-T		0.57	−0.19 cde	0.17 b	−0.22	0.11	0.04	−0.50	0.43 ab	−0.40	−0.25	0.31	−0.05	0.27	0.31	0.32
SSS-T		1.31	1.12 a	−0.28 b	0.75	−0.08	−1.28	−0.04	−0.02 bc	−0.15	0.59	0.60	0.24	0.19	0.37	0.28
SEM		0.30	0.39	0.55	0.69	0.41	0.23	0.47	0.37	0.48	0.50	0.43	0.41	0.46	0.37	0.41

^†^ CCC, continuous corn; Cs, corn phase of the corn-soybean rotation; Sc, soybean phase of the corn-soybean rotation; SSS, continuous soybean. ^‡^ NT, no-till; T, chisel tillage. ^•^ CCC-NT, continuous corn, and no-till; CCC-T, continuous corn, and chisel tillage; Cs-NT, corn phase of the corn-soybean rotation and no-till; Cs-T, corn phase of the corn-soybean rotation and chisel tillage; Sc-NT, soybean phase of the corn-soybean rotation and no-till; Sc-T, soybean phase of the corn-soybean rotation and chisel tillage; SSS-NT, continuous soybean and no-till; SSS-T, continuous soybean and chisel tillage.

## Data Availability

The data presented in this study is available in the Mendeley Database at doi:10.17632/vjn8bf5fvy.2 (Available after 10 August 2021) [117].

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
