# Peer review of "Soil Microbial Indicators within Rotations and Tillage Systems"

_microorganisms, 2021, doi:10.3390/microorganisms9061244_

Round 1

Reviewer 1 Report

The materials and methods are overly long and should be shortened.

The results are overly long and should be shortened

Combined shortened results with the discussion

Pair down the figures and tables.

Author Response

The materials and methods are overly long and should be shortened.

The results are overly long and should be shortened

Combined shortened results with the discussion

Pair down the figures and tables.

Author’s response: Thank you for reviewing our manuscript. We believe your comments and suggestions have improved the quality of our work. We have significantly reduced the conclusion and clarified many other aspects of the manuscript in addressing other reviewer's specific comments. We were a little unsure of how to address some of your comments as they were in opposition to the other two reviewers. We thank you for your comments and are grateful for your time.

Reviewer 2 Report

The manuscript "Soil microbial indicators within rotations and tillage systems" is a very hight quality papers and the scopus is very clear. The presentation of results are very nice. 

I suggest to accept as it.

Author Response

The manuscript "Soil microbial indicators within rotations and tillage systems" is a very high quality papers and the scopus is very clear. The presentation of results are very nice.

I suggest to accept as it.

Author’s response: Thank you for your very positive comments and your time in reviewing this manuscript.

Reviewer 3 Report

Review Report

Manuscript No: microorganisms-1247538

Title: Soil microbial indicators within rotations and tillage systems

Abstract: Quantitative information regarding the changes in pH and SOM will be helpful in the abstract section.

Line 28-29: Reference is required. Please add.

Line 69: What are those kingdoms? Please specify.

Line 72-73: Please add the reference for “Indicator microbes usually refer to an ASV that explains variability in a dataset”

Line 86: Soil microbiome (This refer to both microorganisms and metabolic products synthesized.

Line 90: Please improve the sentence. It should be started with “Experiment was conducted…………”

Line 102: Why there was difference in the application of N fertilizer. These variations may also be the plausible reason for changes in microbial community structure. Please explain.

A brief plot design may be provided in Section 2.1 for better understanding to readers.

Line 106: “Postharvest” rather than “post-harvest”

Line 117-119: To which base the letter “Y, M, S, W, V” refers in primer sequence?

Line 150-151: This refers to column or row, must be specified.

Line 207: Why “Bacteria” rather than “bacteria”?. Same points are also for Line 213 “ Fungi”, Line 219 “Archaea”

Discussion: In my opinion, term microbiome is much broad and includes microorganisms together with the metabolites synthesized by them. The authors must be specific whether they wish to shed light on the results pertaining to microbiome or microbial indicators. Please clarify and amend suitable changes throughout the manuscript.

Conclusion: Please reduce the length of conclusion. The repetitions of results are not required in this section.

The work has presented the good information regarding the changes in microbial community during different agricultural practices. Analysis of microbiological changes under varying agro-practices may be a possible route for assessing and managing the crop productivity. This work is of utmost agricultural importance and must be accepted for publication.

Author Response

Author’s response: Thank you for your time and effort reviewing our manuscript. We were able to incorporate your comments and suggestions. We no doubt believe that your comments improved the manuscript.

Abstract: Quantitative information regarding the changes in pH and SOM will be helpful in the abstract section.

Author’s response: Agreed. We have added a line describing those changes quantitatively.

Line 28-29: Reference is required. Please add.

Author’s response: Agreed. We have added references to bolster our point.

Line 69: What are those kingdoms? Please specify.

Author’s response: Agreed. We added the kingdoms of bacteria, fungi, and archaea to clarify.

Line 72-73: Please add the reference for “Indicator microbes usually refer to an ASV that explains variability in a dataset”

Author’s response: Agreed. We added a reference to clarify.

Line 86: Soil microbiome (This refer to both microorganisms and metabolic products synthesized.

Author’s response: Agreed. We changed “microbiome” to “soil microorganisms” to be specific.

Line 90: Please improve the sentence. It should be started with “Experiment was conducted…………”

Author’s response: Agreed. We changed the sentence to “the experiment was conducted….”.

Line 102: Why there was difference in the application of N fertilizer. These variations may also be the plausible reason for changes in microbial community structure. Please explain.

Author’s response: We agree that the change in fertilizer rate is confounded with the crop rotation and we make no claims on N rates shifting soil indicator species in the manuscript. The typical management in these systems is to apply more N to monoculture corn compared to rotated corn in the attempt to not diminish yields. The N rates were suggestions based on the Illinois Agronomy Handbook and we followed those protocols.

A brief plot design may be provided in Section 2.1 for better understanding to readers.

Author’s response: Thank you for your suggestion, while we agree in theory, we feel that the graphical abstract does that already.

Line 106: “Postharvest” rather than “post-harvest”

Author’s response: Agreed. We changed to postharvest.

Line 117-119: To which base the letter “Y, M, S, W, V” refers in primer sequence?

Author’s response: Those are changes to historical primers. The references following each primer sequence describes what those mean.

Line 150-151: This refers to column or row, must be specified.

Author’s response: Agreed. We have specified the significance is only within columns and for each taxon group.

Line 207: Why “Bacteria” rather than “bacteria”?. Same points are also for Line 213 “ Fungi”, Line 219 “Archaea”

Author’s response: Agreed. We have changed those typos. Thank you for catching that.

Discussion: In my opinion, term microbiome is much broad and includes microorganisms together with the metabolites synthesized by them. The authors must be specific whether they wish to shed light on the results pertaining to microbiome or microbial indicators. Please clarify and amend suitable changes throughout the manuscript.

Author’s response: Agreed. We have removed the inappropriate references to the microbiome to be clearer and not cause any misunderstandings. Thank you for bringing this to our attention.

Conclusion: Please reduce the length of conclusion. The repetitions of results are not required in this section.

Author’s response: Agreed. We have reduced the length and removed the redundancy issues. Thank you for your suggestion.

Round 2

Reviewer 3 Report

The authors responded to all my remarks.